# Optimal Management of Patients with Severe Coronary Artery Disease Following Multidisciplinary Heart Team Approach—Insights from Tertiary Cardiovascular Care Center

**DOI:** 10.3390/ijerph19073933

**Published:** 2022-03-25

**Authors:** Szymon Jonik, Michał Marchel, Ewa Pędzich-Placha, Arkadiusz Pietrasik, Adam Rdzanek, Zenon Huczek, Janusz Kochman, Monika Budnik, Radosław Piątkowski, Piotr Scisło, Paweł Czub, Radosław Wilimski, Jakub Maksym, Marcin Grabowski, Grzegorz Opolski, Tomasz Mazurek

**Affiliations:** 1Department of Cardiology, Medical University of Warsaw, Banacha 1a Str., 01-267 Warsaw, Poland; szymonjonik.wum@gmail.com (S.J.); michal.marchel@wum.edu.pl (M.M.); ewa.pedzich@wp.pl (E.P.-P.); apietrasik@tlen.pl (A.P.); ardza@wp.pl (A.R.); zhuczek@wp.pl (Z.H.); jkochman@tlen.pl (J.K.); moni2911@interia.pl (M.B.); radekp1@gmail.com (R.P.); piotr.scislo@gmail.com (P.S.); jakub.maksym@gmail.com (J.M.); marcin.grabowski@wum.edu.pl (M.G.); grzegorz.opolski@wum.edu.pl (G.O.); 2Department of Cardiac Surgery, Medical University of Warsaw, Banacha 1a Str., 01-267 Warsaw, Poland; pczub73@gmail.com (P.C.); rwilimski@gmail.com (R.W.)

**Keywords:** multidisciplinary heart team, multivessel coronary artery disease, coronary artery bypass grafting, percutaneous coronary intervention, optimal medical therapy

## Abstract

Background: The purpose of this retrospective study was to investigate outcomes of patients with severe coronary artery disease (CAD) after implementing various treatment strategies following multidisciplinary Heart Team (MHT) discussion. Methods Primary and secondary endpoints and quality of life during a mean (SD) follow-up of 37 (14) months of patients with severe CAD (three-vessel [3-VD] or/and left main [LM] disease) qualified after MHT discussion to optimal medical treatment (OMT) alone, OMT and coronary artery bypass grafting (CABG), or OMT and percutaneous coronary intervention (PCI) were evaluated. As the primary endpoint, major adverse cardiac or cerebrovascular events (MACCE) (i.e., death from any cause, stroke, myocardial infarction, or repeat/need for revascularization) were considered. Result: From 2016 to 2019, 176 MHT meetings were held, and a total of 1286 participants with severe CAD and completely implemented MHT decisions (OMT, CABG, or PCI for 251, 356, and 679 patients, respectively) were included. The occurrence of the primary endpoint was significantly increased in OMT-group (154 (61.4%) vs. CABG and PCI groups—110 (30.9%) and 302 (44.5%) patients, respectively (*p* < 0.05). For interventional strategies only—CABG was associated with reduced rates of MACCE and repeat revascularization, while the superiority of PCI for stroke and disabling stroke was observed (*p* < 0.05). The general health status assessed at the end of the follow-up was significantly better for patients who underwent CABG or PCI than in the OMT group (*p* < 0.05). Conclusions: In this real-life study, we presented a single-center experience of providing optimal medical care for patients with severe CAD following MHT discussion.

## 1. Introduction

Coronary artery disease (CAD), consisting of acute coronary syndromes (ACS) and chronic coronary syndromes (CCS), is a group of clinical diagnoses related to myocardial ischemia, including the manifestation of atherosclerosis in coronary arteries, and is the most common cause of death of over 30% of people older than 35 years [1]. CAD affected 197 million people in 2019 and was generally responsible for 9.14 mil deaths [2,3]. Although age-standardized prevalence and morbidity from CAD per 100,000 persons decreased between 1980 and 2019, especially in developed countries [4,5], it remains the most serious medical problem of the modern world, significantly reducing life expectancy and quality. With a growing number of therapeutic options such as less invasive methods of cardiac surgery, the enormous development of percutaneous methods, and the availability of new drugs improving survival in CAD patients, the idea of a multidisciplinary heart team (MHT) to manage individuals with complex diseases has been implemented and still plays a central concept in the real-life care of patients with CAD (class I recommendation in European and American guidelines) [6,7,8,9,10]. Recently, the results of the ISCHEMIA trial demonstrated that for stable CAD patients without significant left main stenosis (LMS) and moderate to severe ischemia, initial conservative strategy (optimal medical therapy (OMT)) is non-inferior to invasive methods (coronary artery bypass grafting (CABG) or percutaneous coronary intervention (PCI)) regarding the risk of ischemic cardiovascular (CV) events or death from any cause [11]. Therefore, it prompts us to seriously consider which treatment will be the most beneficial for CAD patients and whether invasive always means better, making the decision-making process difficult for individuals. However, though the idea of MHT is generally adopted in medical society, an unsatisfactory consensus on how MHT should cooperate, what desired goals are established, and most importantly, the long-term results of MHT decisions, implementation, and patients’ quality of life are still poorly investigated. Some studies address the association of MHT cooperation and MVD-patients evaluation [12,13,14,15,16,17,18,19,20,21]. However, among them, there are very few randomized trials supporting this approach since studies describe outcomes of a multidisciplinary approach without a comparator. The lack of clearly expressed data calls for more evidence investigating basic MHT influence on CAD-patients management and thus CV outcomes. In this retrospective study, we try to evaluate MVD-patients management, long-term outcomes, and quality of life following implementation of MHT decisions in daily clinical practice of tertiary cardiovascular care centers. We believe that the presented results will support and emphasize the evidence-based role of MHT in the decision-making process for MVD-patients and honor the MHT concept as fundamental for linking evidence-based medicine and a multidisciplinary approach for implementing an optimal treatment among an aging and multi-burdened demographic and the rapidly evolving field of cardiovascular and pharmacological medicine.

## 2. Methods and Study Design

This single-center observational study was conducted in the 1st Department of Cardiology, Medical University of Warsaw, a large tertiary cardiovascular care center in Poland. A total of 1509 patients consulted for CAD during 176 MHT meetings in 2016–2019 were enrolled in this retrospective study. The inclusion criteria were: aged ≥ 18 years and complete clinical, echocardiographic, and angiographic characteristics. The angiographic inclusion criterium for final analysis was severe CAD, defined as three-vessel disease (3-VD) and/or left main (LM) disease. The exclusion criteria included the following: pregnancy/lactation, disseminated neoplastic process, life expectancy < 1 year, lack of informed, written consent. All of the patients were evaluated in a weekly meeting by an MHT composed of at least 4 specialists: interventional cardiologist, cardiac surgeon, clinical cardiologist, and non-invasive imaging specialist (often also an anesthesiologist, intensive care specialist, radiologist, and neurologist) and qualified after MHT discussion and unanimous consent of all participating physicians to one of three main strategies: OMT alone; OMT and CABG; or OMT and PCI. To not delay the treatment for acute patients, they were discussed firstly during the meetings, and decisions were implemented immediately. Sequentially, 122 (8.1%) patients presented during MHT meetings were excluded from further analysis due to one vessel disease (1-VD) or two-vessel disease (2-VD) without LM involvement, thus not meeting angiographic criteria for severe CAD or qualification for combined surgery (CABG) and mitral and/or aortic valve surgery—85 and 37 patients, respectively. Sequentially, out of 1387 (91.9%) patients, 128 (9.2%) re-discussed cases qualified for OMT, CABG, or PCI, a total number of 101 (7.3%) participants were excluded due to no consent between MHT and patient preference, loss of follow-up, or death before implementation—58, 29, and 14 patients, respectively. Ultimately, in the final study, 1286 (85.2%) patients with completely implemented MHT decisions (OMT, CABG, PCI—251, 356, 679 patients, respectively) were included. OMT was defined as using drugs with proven impact on increased survival or providing optimal reduction of the signs and symptoms associated with CAD. The severity of CAD or Heart Failure (HF) symptoms were assessed using Canadian Cardiovascular Society (CCS) or New York Heart Association (NYHA) classifications, respectively; chronic kidney disease (CKD) defined as glomerular filtration rate (GFR) <60 mL/min/1.73 m^2^ (mL per min per 1.73 square m), severe pulmonary arterial hypertension (PAH) as pulmonary artery systolic pressure (PASP) >55 mmHg, anemia as hemoglobin level <12 g/dL for women and <14 g/dL for men (g/dL—gram/decilitre), and cancer—active or up to 5 years back. The LM disease was defined as LM stenosis >50%. As the primary endpoint, a major adverse cardiac or cerebrovascular event (MACCE) (i.e., death from any cause, stroke, myocardial infarction, or repeat/need for revascularization) at the end of follow-up (EOF) was considered, while assessed independently the components of MACCE, composite of all-cause death, MI or stroke, and cardiovascular (CV) death, disabling stroke, in-hospital mortality, and for interventional strategies—graft occlusion or stent thrombosis were determined as secondary endpoints. All participants were observed for the occurrence of endpoints with a mean follow-up (SD) of 37 (14) months. The main outline of the study is presented in Figure 1. Additionally, general health status using the short-form (SF)-36 questionnaire (totally and separately for physical component summary [PCS] and mental component summary [MCS]) before CABG, PCI, and MHT discussions (for patients qualified for OMT) and at the EOF for all living participants (31 December 2020) was assessed. We have not yet obtained ethical/institutional review board (IRB) approval for our research. However, due to the observational nature of the study, in accordance with applicable regulation, it is not required.

## 3. Statistical Analysis

The PQStat software (version 1.6.6, PQStat, Poznań, Poland) was used for statistical analysis. The normality of distribution for continuous variables was confirmed with the Shapiro–Wilk test. Categorical data were expressed as counts and percentages, while continuous data were presented as mean (SD). The comparison between groups of patients qualified for individual treatment strategies was performed using the chi-square test, and the statistical analysis was executed using 1-way analysis of variance (ANOVA). To compare the outcomes for all strategies, the hazard ratios (HRs) with 95% confidence intervals (95% CI) were calculated. Time to event analysis was performed using Kaplan-Meier curves. All *p* values (*p*) were given to at least 2-sided, and *p* values lower than 0.05 were considered statistically significant.

## 4. Results

### 4.1. Study Population and Baseline Characteristics

From January 2016 to December 2019, 176 MHT meetings were held and total of 1286 patients with severe CAD (3-VD or/and LM disease) meeting inclusion and exclusion criteria with completely implemented MHT decisions (963 (74.9%) male, age (years, mean (SD)) = 69.0 (10.0), BMI (Body Mass Index) (kg/m^2^ (kilogram per square meter), mean (SD)) = 27.9 (3.5), EuroSCORE II (European System for Cardiac Operative Risk Evaluation II) (%, mean (SD)) = 5.5 (1.7), STS (Society of Thoracic Surgeons) score (%, mean (SD)) = 3.5 (1.2) and given co-morbidities) were followed up. Nearly 41% of participants presented with myocardial infarction with or without ST-segment elevation or unstable angina, 3.4% with cardiogenic shock, while the rest had chronic symptoms of CAD. Regarding statistically significant differences between CABG, PCI, and OMT groups, patients who qualified for OMT were significantly older and more frail. They presented more often with HF, severe left ventricle (LV) dysfunction (EF < 30%), increased left ventricle end-diastolic diameter (LVEDD) and NYHA III-IV class, were often burdened with atrial fibrillation (AF), CKD, anemia, severe PH, cancer, and with the highest perioperative risk of intervention assessed both by EuroSCORE II and STS than those with implemented CABG or PCI (*p* < 0.01). For interventional strategies, the mean delay time from MHT decision to implementation was significantly longer for CABG-patients than in PCI-group (mean (SD): 8.4 (1.3) vs. 4.2 (0.9) days, respectively; *p* < 0.01). Regarding angiographic parameters, participants qualified for CABG and PCI had more coronary lesions affected and often LM stenosis than those from the OMT group (*p* < 0.01). Complete revascularization was achieved more frequently in CABG-group than in PCI-patients (65.4% vs. 58.5%; *p* < 0.01). Baseline clinical characteristics (overall and by groups) in detail are presented in Table 1.

### 4.2. Medications at Discharge

Optimal medical therapy was administrated for all participants, most frequently using aspirin, P2Y_12_ inhibitor, statin, ACE (Angiotensin-Converting Enzyme) inhibitor, and beta-blocker in the PCI-group, VKA (Vitamin K Antagonist) in the CABG-group, and NOAC (Novel Oral Anti Coagulants) loop diuretic, aldosterone antagonist, and amiodarone in the OMT-group (*p* < 0.05). The discharged medications are presented in Table 2.

### 4.3. Endpoints

The occurrence of the primary endpoint was statistically the most frequent in the OMT group (154 patients (61.4%)), while in CABG and PCI groups—110 (30.9%) and 302 (44.5%) patients, respectively (*p* < 0.05). Excluding in-hospital mortality, CABG and PCI were significantly superior to OMT for other secondary endpoints (*p* < 0.05). Considering the endpoints for interventional strategies only, CABG was associated with reduced risk of MACCE and repeat revascularization, while the superiority of PCI for stroke and disabling stroke was observed (*p* < 0.05). However, no statistically significant differences between CABG and PCI concerning the occurrence of the composite of all-cause mortality, stroke, or MI, and all-cause mortality, CV death, MI and graft occlusion, or stent thrombosis were observed (*p* > 0.05). Postprocedural hospital stay was significantly longer for CABG-patients than in PCI-group (mean (SD): (9.9 (1.4) vs. 4.3 (0.7) days, respectively; *p* < 0.01). The endpoints comparing CABG, PCI, and OMT are detailed in Table 3. The Kaplan-Meier curves comparing all strategies for primary and secondary endpoints are presented in Figure 2.

### 4.4. Quality of Life

General health status assessed before implementing MHT decisions (PCS, MCS, and total) did not statistically differ between treatment strategies (*p* > 0.05). At the EOF, the results of PCS, MCS, and the total for all living participants were significantly the lowest for CABG, then for PCI, and highest for OMT-group (*p* < 0.05) as detailed in Table 4. According to the Polish version of the questionnaire, with a maximum of 103 points for PCS and 68 points for MCS (171 points total), the highest point value means the lowest quality of life assessment, while the lowest point value indicates the highest level of quality of life.

### 4.5. Logistic Regression Analysis

Moreover, to determine the factors (from baseline clinical characteristics, echocardiographic, and angiographic parameters) independently related to increased occurrence of primary endpoint and decreased survival rate, we have performed multivariable, multinominal logistic regression analysis involving the overall cohort of patients with severe CAD following MHT discussion and decisions implementation.

Our analysis revealed that (independent, of final treatment strategies—CABG + OMT, PCI + OMT, OMT) age, cardiogenic shock on admission, LV dysfunction (EF < 30 %), NYHA class III-IV, diabetes requiring insulin, PAD, CKD, anaemia, COPD, severe PH, cancer, frailty, LM disease, CTO and EuroSCORE II, were independent factors associated with increased occurrence of primary endpoint in long term follow-up (*p* = 0.02, <0.01, <0.01, 0.03, 0.04, 0.02, <0.01, <0.01, <0.01, <0.01, <0.01, <0.01, 0.03, 0.01, <0.01, respectively).

Independent factors negatively associated with 3-year survival (mean follow-up (SD) = 37 (14) months) were the same as for primary endpoint, but also male gender, previous stroke/TIA and carotid AD (*p* = 0.04, <0.01, 0.02, respectively).

## 5. Discussion

In this retrospective study examining patients with severe CAD who underwent MHT evaluation and were then treated in accordance with MHT decisions, the main findings were as follows: (1) patients presented during MHT consultations had a mean age of 69, high EuroSCORE II, STS and SYNTAX score, nearly one-third had diabetes and more than 25% had severe LV dysfunction; (2) patients qualified for OMT were significantly older, much burdened with comorbidities and frailty, with increased LVEDD and higher risk of intervention (assessed both by EuroSCORE II and STS score) compared with those qualified for CABG or PCI; (3) participants had increased odds of receiving CABG if they were younger, male and with LM disease, while less often had HF, severe symptoms of HF (NYHA III-IV), or severe LV dysfunction (EF < 30%), history of AF, previous MI or CABG, CKD, anaemia, severe PH, cancer, or more often presented with cardiogenic shock and with the most severe symptoms of CAD—clinical (CCS III-IV) and angiographic (SYNTAX score); (4) those with the highest BMI and the largest number of lesions affected had received PCI; (5) excluding in-hospital mortality, the occurrence of primary and secondary endpoints were significantly increased in OMT-group; (6) comparing interventional strategies—rates of MACCE were significantly higher in the PCI-group, in large part because of an increased rate of repeat revascularization (24.3% vs. 10.7%; *p* < 0.05); (7) although stroke and disabling stroke were more frequent in CABG-patients and PCI-patients had higher rates of repeat revascularization, no difference in hard clinical endpoints as composite of all-cause mortality, stroke, or MI, and all-cause mortality, in-hospital mortality, CV death, MI, graft occlusion, or stent thrombosis were observed.

Although risk assessment appears to be a decisive component in the relevant preprocedural selection of the optimal management strategy for patients with severe CAD, there are limitations to the scoring systems used to estimate the risk of adverse outcomes, and many other conditions should be evaluated to properly choose the best treatment option. Hence, the decision cannot be attributed to individuals but to a group of specialists. In our self-experience, the MHT approach was suitable and effective for individual-specific decision-making in patients with severe CAD. One of the main challenging problems for MHT is the management of critically ill or hemodynamically unstable patients who need an urgent decision. That is why only 3.4% of participants we discussed had presented with cardiogenic shock. Which is worth noting, more than a third of patients (40.9%) were admitted due to unstable angina or MI and—despite the fact they need urgent management—were carefully discussed by MHT. Hence, we supposed that a majority of the medical community believe that MHT will provide the most effective treatment for their patients. We observed that patients qualified for OMT were the oldest: we think there are basically two reasons for this phenomenon—firstly, in old age, people are afraid of invasive strategy and prefer pills; secondly—for elderly (mainly due to high risk)—cardiologists are more careful about intervention. This finding may reflect that we pay attention to many factors not included in commonly used risk scales—for example frailty, poor mobility, and patient preferences, which give our observations and results real-life clinical value. Considering interventional treatment, for younger patients but with significant LM disease, MHT tended to choose CABG, while for those with severe HF, LV dysfunction, and with more comorbidities, we prefer PCI. These results reflect recommendations and general perception of the medical community, which support better long-term outcomes associated with CABG and greater safety of PCI for highly burdened patients. Interestingly, we did not observe a trend in choosing a treatment strategy for diabetic patients, and even statistically insignificant, but a greater percentage of these participants received PCI or OMT. This is somewhat different from the current guidelines for myocardial revascularization, in which for patients with LM disease or 3-VD and high SYNTAX score, CABG is recommended strategy [22]. The discrepancy may be explained by the fact that the cohort of patients consulted by our internal MHT is not representative of the overall population of CAD individuals, and we attach great importance to the preferences of patients who are often afraid of open surgery.

Through our study, we would highlight the need for research to determine the MHT definition and range of functioning by which it can be assessed to advance our comprehension of the optimal care model for CAD patients. The development of new therapeutic options and now and then changes in evidence-based cardiology have added to the complexity of medical decision-making. Importantly, since we have started our research, there have been major technical and procedural advancements in interventional cardiology. These include physiology-based revascularization methods, such as the hybrid use of instantaneous wave-free ratio (iFR) and fractional flow reserve (FFR), newer generations drug-eluting stents, intravascular ultrasound (IVUS)-guided optimization of stent deployment, improved contemporary chronic total occlusion (CTO) revascularization techniques, and improved guideline-directed medical therapy. Therefore, in each subsequent year of the study, we referred more and more patients to PCI.

Considering endpoints for all strategies, the OMT group had demonstrated the highest frequency of most MACCE (excluding in-hospital mortality and repeat/need for revascularization), mostly due to old age, multiple comorbidities, and frailty. For interventional strategies only, beyond similar outcomes (for in-hospital mortality, all-cause mortality, CV death, MI and graft occlusion or stent thrombosis)—CABG was associated with a lower occurrence of MACCE and reduced need for repeat revascularization, while the superiority of PCI over CABG for stroke and disabling stroke was observed. Although our study is only single-center and retrospective, its outcomes are consistent with an important multi-center randomized trial for CAD-patients (SYNTAX) [12], in which rates of MACCE at 12 months were significantly higher for the PCI group (17.8%, vs. 12.4% for CABG; *p* = 0.002), mainly due to an increased rate of repeat revascularization (13.5% vs. 5.9%, *p* < 0.001) and at 1 year, the rates of death and MI were similar between PCI and CABG, while the incidence of stroke was significantly frequent in CABG group.

In our opinion, only the cooperation of MHT (where the risk assessment is only a component) provides complex decision-making with an appraisal of factors not routinely included in risk algorithms, which is the best to reflect the circumstances of real-world clinical practice. More importantly, more clinical trials comparing treatment options for CAD patients mainly focus on interventional strategy and neglect the long-term outcomes and quality of life for patients enrolled in conservative management after MHT evaluation.

It seems to be essential to discuss some important studies in which MHT was involved in the selection of appropriate CAD patients and the multifactorial decision-making process. The very first such study, with the MHT conception touched upon, was SYNTAX (outcomes presented above) [12], in which each clinical case and angiogram was reviewed by a team consisting of an interventionist and a surgeon. After a consensus agreement, the decision for which procedure or procedures the patient may be eligible for was made.

Subsequently, the first review article by Head SJ et al. raised the issue of adjusting the MHT concept with available diagnostic evidence, patient preferences, local expertise, and constantly changing recommendations to provide the most beneficial outcomes for CAD patients [13]. After that, in 2014, 5-year outcomes of patients with 3-VD (*n* = 1095) treated with CABG or PCI using the first-generation paclitaxel-eluting stents from SYNTAX trial were revealed, resulting in significantly lower rates of death, MI, and repeat revascularization in the CABG cohort, while stroke rates were similar [14]. Furthermore, Bonzel T et al., and then Abdulrahman M, et al. stressed the impact of the structured MHT approach and the hierarchy of the participating physicians for clinical decision-making associated with improved outcomes [15,16]. In another study by Collet C et al., separate MHTs composed of an interventional cardiologist, a cardiac surgeon, and a radiologist were randomized to assess the CAD with either coronary computed tomography angiography (CTA) or conventional angiography in 223 patients with de novo LM stenosis or 3-VD. MHT compliance in assessing patients’ qualification for PCI or CABG procedures (the primary endpoint) was very high and amounted to approximately 93%. Cohen’s Kappa coefficient was 0.82, indicating almost complete agreement between the two teams. Additional data from the non-invasive assessment of FFR CTA (a secondary endpoint) changed the decision of MHT in 14 patients (7%), of which in 13 patients it meant a re-qualification from surgery to percutaneous treatment [17].

Afterward, some observational studies evaluated the MHT approach for CAD patients in single-center experiences [18,19,20]. In the study by Patterson T et al. from 2012 to 2013, 51 MHT meetings were held, and a total of 366 cases were discussed providing MHT recommendation and outcome of CABG + OMT, PCI + OMT, and OMT alone in 27.9%, 34.7%, and 37.4% of cases, respectively. The 3-year survival rate was 60.8%, 84.3%, and 90.2% in the OMT, OMT + PCI, and 90.2% in the OMT + CABG cohorts, respectively. Medical therapy was associated with a 4.5-fold increased risk of mortality compared with CABG and PCI (HR: 4.588; 95% CI: 2.333–9.021; *p* < 0.001) [18]. Dominiques CT et al. presented assigning nearly 1000 patients to CABG, PCI, OMT, or additional diagnostic methods depending on the number of affected coronary vessels, MHT decisions, and patients’ preferences, emphasizing that the final MHT recommendation was largely in accordance with clinical guidelines [19]. In another research, out of the 166 patients discussed at MHT meetings, 79 (47.6%) underwent PCI, 49 (29.5%) underwent CABG, 1 (0.6%) underwent hybrid revascularization, and 34 (20.5%) were treated with OMT only. Among 129 patients who underwent revascularization (PCI or CABG), the in-hospital and 30-day mortality was 3.9% and 4.8%, respectively, while there were no trends in recommendations for CABG, PCI, or OMT by SYNTAX score tertiles [20].

Very recently, the SYNTAX II strategy of incorporating both clinical and anatomical variables into MHT decisions to guide myocardial revascularization led to better 5-year clinical outcomes in comparison with the SYNTAX trial, which evaluated anatomic factors only [21]. A total of 454 patients were included and compared with 315 patients from the pre-defined SYNTAX PCI group and 334 patients from the pre-defined SYNTAX CABG cohort. At 5 years, MACCE (composite of all-cause death, stroke, any MI, and any revascularization) occurred in 21.5% of SYNTAX II patients, which was significantly lower than the 36.4% MACCE rate in the SYNTAX PCI group (HR: 0.54; 95% CI: 0.41–0.71; *p* < 0.001). All MACCE components, except stroke, were significantly lower in SYNTAX II PCI patients [21]. This study proves the relevance of MHT management—taking into account that clinical parameters through decision-shared making resulted in better clinical outcomes for SYNTAX II patients than a matched PCI group from SYNTAX.

There are several methodological strengths in this study that reinforce the validity of the obtained results: all-comer nature, retrospective enrolment, systematic and meticulous patient assessment, complete mean 37-months clinical follow-up, assessment of the quality of life, and the use of standardized definitions and endpoints for clinical outcomes support the universal value of these data. Moreover, quite a large group of patients (as regards the conditions of single-center study) and a long follow-up is sufficient to determine with high probability that decisions of our MHT are adequate and consistent with clinical practice. Furthermore, properly selected endpoints, clearly reflecting the most common and serious complications of CAD-interventional treatment, prove the translatability of the obtained results on the proper functioning of MHT.

## 6. Conclusions

In our study, we presented how the MHT approach and decisions affect the prognosis and quality of life of patients with severe CAD, demonstrating that those qualified by our internal MHT for interventional strategy achieved greater benefits in both endpoints and long-term quality of life as compared to pharmacological treatment only. It should be especially emphasized that for CAD patients, choosing the best treatment method should never be individual. The results of this study require further confirmation in longer follow-up or multi-center studies and registers, but our initial findings may establish a cornerstone for the future providing establishment of MHT role both in clinical practice and guidelines for CAD management.

### Limitations

This study should be interpreted given the following limitations. The main one is its retrospective, non-randomized character, and single-center design. Above that, the decision-making process must be assigned to our individual MHT cooperation and cannot be considered a general one. Additionally, proper and regular use of drugs by patients often remains a matter of trust; hence it is difficult to determine the credibility of the endpoints in the OMT group. Moreover, patients with non-implemented decisions were not included in the final analysis, so we do not have their follow-up data. Participants were not matched; hence the comparison of groups should be considered with caution. Individuals qualified for interventional strategies differ significantly in some clinical parameters. Thus, the obtained outcomes cannot contribute to formulating far-reaching and unquestionable conclusions.

## Figures and Tables

**Figure 1 ijerph-19-03933-f001:**
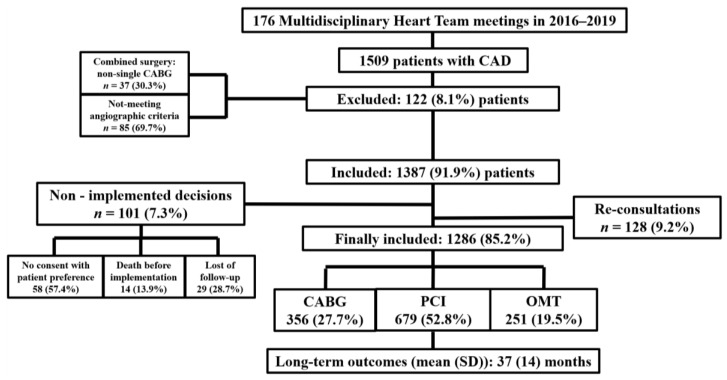
Study design. CAD—coronary artery disease, CABG—coronary artery bypass grafting, PCI—percutaneous coronary intervention, OMT—optimal medical therapy.

**Figure 2 ijerph-19-03933-f002:**
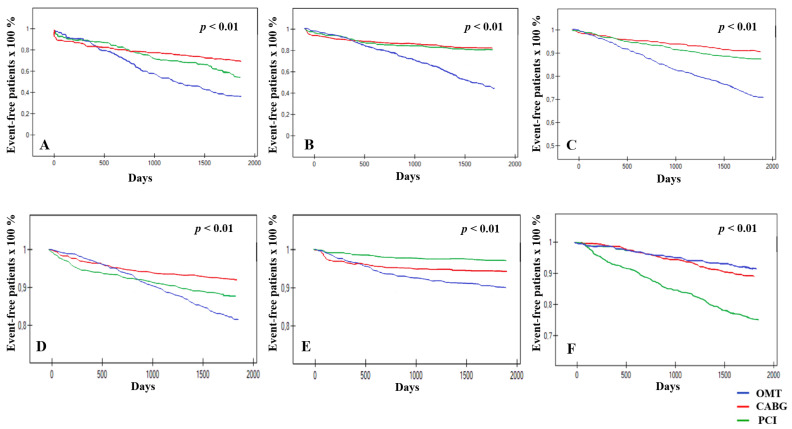
The Kaplan-Meier curves for endpoints. (**A**)—MACCE (death from any cause, stroke, myocardial infarction, or repeat/need for revascularization), (**B**)—all-cause mortality, stroke or myocardial infarction (MI), (**C**)—all-cause mortality, (**D**)—MI, (**E**)—stroke, (**F**)—repeat/need for revascularization.

**Table 1 ijerph-19-03933-t001:** Baseline clinical characteristics.

Baseline Characteristic	Overall (1286)	CABG (356)	PCI (679)	OMT (251)	*p*-Value
Age, years; mean (SD)	69.0 (10.0)	66.9 (9.2)	68.8 (10.1)	72.5 (9.9)	<0.01
Gender, male (%)	963 (74.9)	289 (81.2)	495 (72.9)	179 (71.3)	<0.01
BMI, kg/m^2^; mean (SD)	27.9 (3.5)	27.9 (3.3)	28.2 (3.7)	27.2 (3.0)	<0.01
**Clinical Presentation**					
ACS, *n* (%)	526 (40.9)	152 (42.7)	285 (42.0)	89 (35.5)	0.14
Cardiogenic shock, *n* (%)	44 (3.4)	4 (1.1)	29 (4.3)	11 (4.4)	0.02
Heart Failure, *n* (%)	965 (75.0)	236 (66.3)	498 (73.3)	231 (92.0)	<0.01
LV dysfunction (EF < 50%), *n* (%)	1067 (83.0)	289 (81.2)	567 (83.5)	211 (84.1)	0.56
LV dysfunction (EF < 30%), *n* (%)	336 (26.1)	46 (12.9)	147 (21.6)	143 (57.0)	<0.01
LVEDD, cm (SD)	5.8 (1.0)	5.5 (0.9)	5.8 (1.0)	6.2 (1.0)	<0.01
NYHA class III-IV, *n* (%)	441 (34.3)	95 (26.7)	217 (32.0)	129 (51.4)	<0.01
CCS class III-IV, *n* (%)	518 (40.3)	165 (46.3)	266 (39.2)	87 (34.7)	0.01
Diabetes, *n* (%)	393 (30.6)	98 (27.5)	219 (32.3)	76 (30.3)	0.29
Requiring insulin, *n* (%)	131 (10.2)	29 (8.1)	76 (11.2)	26 (10.4)	0.30
Hypertension, *n* (%)	1068 (83.0)	291 (81.7)	577 (85.0)	200 (79.7)	0.12
Previous stroke/TIA, *n* (%)	112 (8.7)	26 (7.3)	60 (8.8)	26 (10.4)	0.42
Atrial fibrillation, *n* (%)	351 (27.3)	67 (18.8)	189 (27.8)	95 (37.8)	<0.01
Previous MI, *n* (%)	615 (47.8)	189 (53.1)	319 (47.0)	107 (42.6)	0.03
Previous PCI, *n* (%)	382 (29.7)	91 (25.6)	217 (32.0)	74 (29.5)	0.10
Previous CABG, *n* (%)	115 (8.9)	20 (5.6)	71 (10.5)	24 (9.6)	0.03
PAD, *n* (%)	78 (6.1)	14 (3.9)	49 (7.2)	15 (6.0)	0.11
Carotid AD, *n* (%)	133 (10.3)	34 (9.6)	77 (11.3)	22 (8.8)	0.44
CKD, *n* (%)	463 (36.0)	66 (18.5)	204 (30.0)	193 (76.9)	<0.01
Anaemia, *n* (%)	428 (33.3)	89 (25.0)	182 (26.8)	157 (62.5)	<0.01
Dyslipidemia, *n* (%)	1028 (79.9)	284 (79.8)	556 (81.9)	188 (74.9)	0.06
COPD, *n* (%)	129 (10.0)	29 (8.1)	70 (10.3)	30 (12.0)	0.29
Severe PH, *n* (%)	125 (9.7)	17 (4.8)	73 (10.8)	35 (13.9)	<0.01
Cancer, *n* (%)	215 (16.72)	22 (6.2)	104 (15.3)	89 (35.5)	<0.01
Current smoking, *n* (%)	234 (18.2)	65 (18.3)	132 (19.4)	37 (14.7)	0.26
Frailty, *n* (%)	261 (20.3)	9 (2.5)	89 (13.1)	163 (64.9)	<0.01
**Angiographic Characteristics**					
No. of lesions, mean (SD)	4.2 (1.5)	4.2 (1.4)	4.3 (1.5)	3.8 (1.4)	<0.01
LM disesase (LM stenosis ≥ 50%), *n* (%)	313 (24.3)	109 (30.6)	158 (23.3)	46 (18,3)	<0.01
Bifurcation, *n* (%)	927 (72.1)	261 (73.3)	491 (72.3)	175 (69.7)	0.61
CTO, *n* (%)	305 (23.7)	77 (21.6)	164 (24.2)	64 (25.5)	0.50
SYNTAX score; mean (SD)	30.2 (6.3)	31.1 (5.9)	29.6 (6.5)	30.6 (6.1)	<0.01
Complete revascularization, *n* (%)	630/1035 (60.9)	233 (65.4)	397 (58.5)	-	0.03
EuroSCORE II, %; mean (SD)	5.5 (1.7)	3.9 (1.1)	6.0 (1.4)	6.5 (1.5)	<0.01
STS score, %; mean (SD)	3.5 (1.2)	2.5 (0.8)	3.9 (1.0)	4.2 (1.1)	<0.01
Time to procedure, days (SD)	5.7 (2.3)	8.4 (1.3)	4.2 (0.9)	-	<0.01

CABG—coronary artery bypass grafting, PCI—percutaneous coronary intervention, OMT—optimal medical therapy, BMI—body mass index, ACS—acute coronary syndrome, LV—left ventricle, LVEDD—left ventricle end-diastolic diameter, Canadian Cardiovascular Society CCS—Canadian Cardiovascular Society, NYHA—New York Heart Association, TIA—transient ischemic attack, MI—myocardial infarction, PAD—peripheral artery disease, AD—artery disease, CKD—chronic kidney disease, COPD—chronic obstructive pulmonary disease, PH—pulmonary hypertension, LM—left main, CTO—chronic total occlusion, SYNTAX—Synergy between PCI with Taxus and Cardiac Surgery, EuroSCORE II—European System for Cardiac Operative Risk Evaluation II, STS—Society of Thoracic Surgeons score.

**Table 2 ijerph-19-03933-t002:** Cardiac-related medications given at discharge.

Medication at Discharge, *n* (%)	Overall (1233)	CABG (339)	PCI (654)	OMT (240)	*p*-Value
Aspirin	1150 (93.3)	302 (89.1)	631 (96.5)	217 (90.4)	<0.01
P2Y_12_ inhibitors	740 (60.0)	71 (20.9)	635 (97.1)	34 (14.2)	<0.01
Vitamin K antagonist (VKA)	68 (5.5)	29 (8.6)	23 (3.5)	16 (6.7)	<0.01
Novel oral anticoagulants (NOAC)	335 (27.2)	52 (15.3)	186 (28.4)	97 (40.4)	<0.01
Statin	1135 (92.1)	294 (86.7)	628 (96.0)	213 (88.8)	<0.01
ACE inhibitor	853 (69.2)	206 (60.8)	479 (73.2)	168 (70.0)	<0.01
Angiotensin II-receptor antagonist	278 (22.5)	85 (25.1)	144 (22.0)	49 (20.4)	0.37
Beta-blocker	952 (77.2)	270 (79.6)	528 (80.7)	154 (64.2)	<0.01
Calcium-channel blocker	294 (23.8)	71 (20.9)	157 (24.0)	66 (27.5)	0.19
Loop diuretic	875 (71.0)	228 (67.3)	434 (66.4)	213 (88.8)	<0.01
Aldosterone antagonist	396 (32.1)	43 (12.7)	187 (28.6)	166 (69.2)	<0.01
Amiodarone	75 (6.1)	27 (8.0)	28 (4.3)	20 (8.3)	0.02

CABG—coronary artery bypass grafting, PCI—percutaneous coronary intervention, OMT—optimal medical therapy, P2Y_12_ inhibitors—clopidogrel, prasugrel, ticagrelor, ACE—angiotensin-converting enzyme.

**Table 3 ijerph-19-03933-t003:** Primary and secondary endpoints.

Endpoints, *n* (%)	CABG (356)	PCI (679)	OMT (251)	*p* Value Overall	CABG vs. PCI HR [95% CI]; *p*	CABG vs. OMT HR [95% CI]; *p*	PCI vs. OMT HR [95% CI]; *p*
**Primary Endpoint—MACCE**	110 (30.9)	302 (44.5)	154 (61.4)	<0.01	<0.01	<0.01	<0.01
**Secondary Endpoints**							
All-cause mortality, stroke, or MI	69 (19.4)	136 (20.0)	139 (55.4)	<0.01	0.80	<0.01	<0.01
All-cause mortality	32 (9.0)	75 (11.0)	72 (28.7)	<0.01	0.30	<0.01	<0.01
CV death	24 (6.7)	64 (9.4)	43 (17.1)	<0.01	0.14	<0.01	<0.01
In-hospital mortality	17 (4.8)	25 (3.7)	11 (4.4)	0.68	0.40	0.82	0.62
MI	29 (8.1)	78 (11.5)	48 (19.1)	<0.01	0.09	<0.01	<0.01
Stroke	21 (5.9)	14 (2.1)	24 (9.6)	<0.01	<0.01	0.09	<0.01
Disabling stroke	13 (3.7)	6 (0.9)	15 (6.0)	<0.01	<0.01	0.18	<0.01
Repeat/need for revascularization	38 (10.7)	165 (24.3)	19 (7.6)	<0.01	<0.01	0.20	<0.01
CABG	9 (2.5)	17 (2.5)	0 (0.0)	0.04	0.98	0.01	0.01
PCI	29 (8.1)	148 (21.8)	19 (7.6)	<0.01	<0.01	0.80	<0.01
Graft occlusion or stent thrombosis	20 (5.6)	40 (5.9)			0.86		
Acute (at ≤1 day)	2 (0.6)	5 (0.7)			0.75		
Subacute (within 2–30 days)	3 (0.8)	13 (1.9)			0.18		
Late (within 31–365 days)	9 (2.5)	8 (1.2)			0.10		
Very late (≥366 days)	6 (1.7)	14 (2.1)			0.68		
Postprocedural hospital stay, days; mean (SD)	9.9 (1.4)	4.3 (0.7)			<0.01		

MACCE—MACCE (death from any cause, stroke, myocardial infarction, or repeat/need for revascularization), CABG—coronary artery bypass grafting, PCI—percutaneous coronary intervention, OMT—optimal medical therapy, CV—cardiovascular, MI—myocardial infarction.

**Table 4 ijerph-19-03933-t004:** The quality of life before and after multidisciplinary Heart Team (MHT) decisions implementation.

Component	CABG (356/324)	PCI (679/604)	OMT (251/179)	*p*-Value
**Physical Component Summary (PCS)**				
Before CABG, PCI, MHT disscusion; mean (SD)	71.1 (18.6)	73.2 (18.4)	73.8 (15.8)	0.12
After CABG, PCI, MHT disscusion—at the end of follow up; mean (SD)	62.0 (17.0)	65.7 (14.3)	75.0 (15.9)	<0.01
**Mental Component Summary (MCS)**				
Before CABG, PCI, MHT disscusion; mean (SD)	51.6 (9.4)	52.0 (9.4)	52.5 (9.0)	0.53
After CABG, PCI, MHT disscusion—at the end of follow up; mean (SD)	43.3 (9.4)	44.9 (9.7)	53.3 (8.9)	<0.01
**Total**				
Before CABG, PCI, MHT disscusion; mean (SD)	122.8 (20.2)	125.2 (20.6)	126.3 (18.7)	0.07
After CABG, PCI, MHT disscusio—at the end of follow up; mean (SD)	105.3 (18.8)	110.7 (16.8)	128.3 (19.5)	<0.01

CABG—coronary artery bypass grafting, PCI—percutaneous coronary intervention, OMT—optimal medical therapy, MHT—multidisciplinary Heart Team.

## Data Availability

The data presented in this study are available on request from the corresponding author.

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
