# Peer review of "Optimal Management of Patients with Severe Coronary Artery Disease following Multidisciplinary Heart Team Approach—Insights from Tertiary Cardiovascular Care Center"

_ijerph, 2022, doi:10.3390/ijerph19073933_

Round 1

Reviewer 1 Report

The Authors conducted a retrospective study including many patients. Between the groups (OMT, CABG, PCI) important hetereogeneous features are present that inficiate the results of the present study. The Authors should use a propensity match score of the three groups or they should analyze the endpoints for each subgroup of patients (diabetic, old patients, frail patients, ecc..) to limit the miscellaneous baseline features. 

Author Response

Dear

We greatly appreciate you having reviewed our manuscript.

Below we presented our responses to your comments.

  1. About your comment: “The Authors conducted a retrospective study including many patients. Between the groups (OMT, CABG, PCI) important hetereogeneous features are present that inficiate the results of the present study. The Authors should use a propensity match score of the three groups or they should analyze the endpoints for each subgroup of patients (diabetic, old patients, frail patients, ecc..) to limit the miscellaneous baseline features.”

Our answer: This is only an observational study, presenting real-life clinical practice of MHT functioning and its involvement in decision-making process for burdened patients in tertiary cardiovascular care center. We agree with your comment that patients differ in some clinical parameters, as we highlighted this in Section “Limitations”: “Participants were not matched, hence comparison of groups should be considered with caution. Individuals qualified for interventional strategies differ significantly in some clinical parameters, hence the obtained outcomes cannot be a contribution to formulating far-reaching and unquestionable conclusions.” Our study is non-randomized and has no control group – we felt that the use of a propensity match score would be unreasonable and therefore we did not this, because our main goal was to present MHT functioning in real-life clinical practice and its impact on decision-making process and patients prognosis, not to compare the three groups assigned to different treatment strategies. 

However, to meet your expectations, we have performed multivariable, multinominal logistic regression analysis to determine the factors related to increased occurrence of primary endpoint and decreased survival rate independently of final treatment strategies approved by MHT.

In Section 4. Results, we have added new paragraph – 4.5. Logistic Regression Analysis, where we have added these findings:

“4.5. Logistic regression analysis

Moreover, to determine the factors (from baseline clinical characteristics, echocardiographic and angiographic parameters) independently related to increased occurrence of primary endpoint and decreased survival rate, we have performed multivariable, multinominal logistic regression analysis in the overall cohort of patients with severe CAD following MHT discussion and decisions implementation.
Our analysis revealed that (independently of final treatment strategies – CABG+OMT, PCI+OMT, OMT) age, cardiogenic shock on admission, LV dysfunction (EF < 30 %), NYHA class III-IV, diabetes requiring insulin, PAD, CKD, anaemia, COPD, severe PH, cancer, frailty, LM disease, CTO and EuroSCORE II were independent factors associated with increased occurrence of primary endpoint in long term follow-up (P=0.02, <0.01, <0.01, 0.03, 0.04, 0.02, <0.01, <0.01, <0.01, <0.01, <0.01, <0.01, 0.03, 0.01, <0.01, respectively). Independent factors negatively associated with 3-year survival (mean follow-up (SD) = 37 (14) months) were the same as for primary endpoint, but also male gender, previous stroke/TIA and carotid AD (P=0.04, <0.01, 0.02, respectively).”

We hope you are satisfied with this answer.

Thank you for taking the time for review this article.

With best regards

Tomasz Mazurek, MD, PhD

1st Department of Cardiology

Medical University of Warsaw,

Banacha 1a Street
01-267 Warsaw, Poland
e-mail: tomaszmazurek.wum@gmail.com
tel.: + 48 22 599 19 58
fax.: + 48 22 599 19 57

Reviewer 2 Report

Correct the tittle - Discussion, MVD has no meening.

Article Optimal Management of Patients with Severe Coronary Artery Disease Following Multidisciplinary Heart Team Discussion. The study included patients with severe coronary artery disease (CAD) undergoing an interventional procedure (CABG and PCI) versus an optimized treatment-only group (OMT). The groups that received interventional treatment had a statistically superior performance compared to OMT. Evidently the most significant limitations were that it was a retrospective, unicentric study in which patients with OMT were unequal, introducing bias in the analysis, with relationships to the severity of the systemic disease and risks. However, the authors highlighted these inequalities that must be taken into account. The biggest highlight of the study was to successfully apply the strategy using a multidisciplinary group to evaluate and implement the best conduct.

Author Response

Dear

We greatly appreciate you having reviewed our manuscript.

Below we presented our responses to your comments.

  1. You have suggested to correct the title, because “discussion” has no meaning. We have changed “discussion” into “approach”.
  2. Also, you have suggested to correct MVD – in the title we have used “severe coronary artery disease” because this study is about patients with three vessel (3-VD) or/and left main (LM) disease, and this is not the same as multivessel coronary artery disease (MVD). So, in this case, we decided that “severe coronary artery disease” would be better than “MVD”.

Moreover, we agree with your comments regarding the limitations of this study; we agree that such limitations exist, as we also noted in the "Limitations" section, while this is an observational study with "real-life" clinical practice and outcomes and such limitations in this type of study cannot be avoided.

We hope you are satisfied with this answer.

Thank you for taking the time for review this article.

With best regards

Tomasz Mazurek, MD, PhD

1st Department of Cardiology

Medical University of Warsaw,

Banacha 1a Street
01-267 Warsaw, Poland
e-mail: tomaszmazurek.wum@gmail.com
tel.: + 48 22 599 19 58
fax.: + 48 22 599 19 57

Reviewer 3 Report

This is a retrospective observational study on the results of decision making by a multidisciplinary heart team in a large number of patients with coronary artery disease. It is interesting in that it reflects the outcome of decision making in routine clinical practice.

Comments/questions to authors:

- Explain the composition and functioning of the heart team: number of physicians involved in the meetings, how they are organized so as not to delay treatment in acute patients, decision making process.

- Clarify the angiographic criteria for patient inclusion. Specifically, does the following sentence mean that only patients with severe 3-vessel disease are included?  Sequentially, 122 (8.1%) patients were excluded from further analysis due to: not meeting angiographic criteria of severe CAD (three vessel disease (3-VD) or/and left main (LM) disease)

- In the conclusions of the study it is stated that: “It should be especially emphasized that for CAD-patients choosing the best treatment method should never be individual and only MHT seems to be a suitable tool to provide a satisfactory outcomes and acceptable quality of life.”. However, the study design does not allow this conclusion to be drawn.

Author Response

Dear

We greatly appreciate you having reviewed our manuscript.

Below we presented our responses to your comments.

  1. About what you asked: “Explain the composition and functioning of the heart team: number of physicians involved in the meetings, how they are organized so as not to delay treatment in acute patients, decision making process.” we have modified and added in the text:

Our answer: Section 2. Methods and study design.

"All of patients were evaluated in a weekly meeting by a MHT composed of at least 4 specialists: interventional cardiologist, cardiac surgeon, clinical cardiologist and non-invasive imaging specialist (often also an anaesthesiologist, intensive care specialist, radiologist and neurologist) and qualified after MHT discussion and an unanimous consent of all participating physicians to one of three main strategies: OMT alone, OMT and CABG or OMT and PCI. In order to not delay the treatment of acute patients, they were discussed firstly during the meetings and the decisions were implemented immediately."

  1. About your comment: “Clarify the angiographic criteria for patient inclusion. Specifically, does the following sentence mean that only patients with severe 3-vessel disease are included? Sequentially, 122 (8.1%) patients were excluded from further analysis due to: not meeting angiographic criteria of severe CAD (three vessel disease (3-VD) or/and left main (LM) disease)”.

Our answer: Section 2. Methods and study design. We have changed it to this “The angiographic inclusion criterium for final analysis was severe CAD defined as 3-VD or/and LM disease. Sequentially, 122 (8.1%) patients presented during MHT meetings were excluded from further analysis due to: one vessel disease (1-VD) or two vessel disease (2-VD) without LM involvement, so not meeting angiographic criteria of severe CAD;….”

  1. About your comment: “In the conclusions of the study it is stated that: “It should be especially emphasized that for CAD-patients choosing the best treatment method should never be individual and only MHT seems to be a suitable tool to provide a satisfactory outcomes and acceptable quality of life.”. However, the study design does not allow this conclusion to be drawn.”

Our answer: Section 6. Conclusions. We agree with your comment. We have removed “…and only MHT seems to be a suitable tool to provide a satisfactory outcomes and acceptable quality of life”. W have changed the last sentence of Section 6. Conclusions into: “The results of this study require further confirmation in longer follow-up or multicentre studies and registers, but our initial findings may state a cornerstone for the future providing establishment of MHT role both in clinical practice and guidelines for CAD management.”

We hope you are satisfied with this answer.

Thank you for taking the time for review this article.

With best regards

Tomasz Mazurek, MD, PhD

1st Department of Cardiology

Medical University of Warsaw,

Banacha 1a Street
01-267 Warsaw, Poland
e-mail: tomaszmazurek.wum@gmail.com
tel.: + 48 22 599 19 58
fax.: + 48 22 599 19 57

Round 2

Reviewer 1 Report

Good job, congratulation

This manuscript is a resubmission of an earlier submission. The following is a list of the peer review reports and author responses from that submission.

Round 1

Reviewer 1 Report

This manuscript is entitled to study the impact of multidisciplinary heart team meetings on the outcome of patients with severe CAD.

Unfortunately this paper has several significant shortcomings. The objective is not clear, presumingly comparing decision making process prior and after introduction of HT. The results do not show any difference as there is not control group/time. The presented results are at least irritating as tripple vessel disease or significant LMS remain surgical domain (according to international guidelines). Almost 20% of patients were treated by OMT. The authors conclude that they present their outcome following HT discussion.

Author Response

Dear

We greatly appreciate you having reviewed our manuscript.

Below we presented our responses to your comments.

  • “Unfortunately this paper has several significant shortcomings. The objective is not clear, presumingly comparing decision making process prior and after introduction of HT. The results do not show any difference as there is not control group/time. The presented results are at least irritating as tripple vessel disease or significant LMS remain surgical domain (according to international guidelines). Almost 20% of patients were treated by OMT. The authors conclude that they present their outcome following HT discussion.”

Our answer: The objective of this study is to assess outcomes and quality of life for patients with severe coronary artery disease (CAD) after implementation of coronary artery bypass grafting (CABG), percutaneous coronary intervention (PCI) or optimal medical therapy (OMT) as recommended by our internal heart team (HT). This is only retrospective study with non-randomized character, we did not compare decision making process prior and after HT evaluation, we also did not have any comparator (control group). This is the limitation of this study and we wrote about it in the section Limitations, page 12, line 394-397: “The main one is its retrospective, non-randomized character and single-centre design. Above that, the decisions-making process must be assigned to our individual HT cooperation and cannot be considered as a general one.” But, answering to your doubts, in the future, we plan to conduct a trial comparing the results of treatment in patients with multivessel CAD with 2 groups: the study group - patients consulted by the HT and the control group - without HT evaluation.

We agree that three-vessel disease (3-VD) or significant left main stenosis (LMS) is primarily the domain of cardiac surgery and we agree that nearly 20 % of individuals with severe CAD qualified only for pharmacological treatment is a quite large percentage. However, this group of patient was qualified only for OMT due to the oldest age, the greatest burden of co-morbidities and frailty, more than 90 % of them had heart failure and nearly 60 % severe left ventricle (LV) dysfunction with increased left ventricular end-diastolic diameter (LVEDD). Moreover, patients from OMT-group had the highest risk of intervention (assessed both by EuroSCORE II and STS score) comparing with those qualified for CABG or PCI. All these factors contributed to their poor prognosis and short predictable life expectancy. In our study patients from OMT-group was disqualified form interventional strategy due to extremely high risk – CABG in these individuals was assessed by our HT as associated with a high risk of intraoperative death and probably would not improve quality of life. These patients had too advanced coronary artery disease and were already too sick to be offered a successful revascularization. Unfortunately, in such a case, surgery would be an unreasonable escalation of therapy, which we considered unethical.

Thank you for taking the time for review this article.

With best regards

Tomasz Mazurek, MD, PhD

1st Department of Cardiology

Medical University of Warsaw,

Banacha 1a Street
01-267 Warsaw, Poland
e-mail: tomaszmazurek.wum@gmail.com
tel.: + 48 22 599 19 58
fax.: + 48 22 599 19 57

Reviewer 2 Report

The background and objectives could be agreed. However, these issues should be done prospectively or all of the baselines should be adjsuted.  

Author Response

Dear

We greatly appreciate you having reviewed our manuscript.

Below we presented our responses to your comments.

  • “The background and objectives could be agreed. However, these issues should be done prospectively or all of the baselines should be adjsuted.”

Our answer:

We agree with this. In section Abstract, page 1, line 14, we have changed “background” into “objectives”. All the baseline characteristics – Table 1 was determined on admission.

Thank you for taking the time for review this article.

With best regards

Tomasz Mazurek, MD, PhD

1st Department of Cardiology

Medical University of Warsaw,

Banacha 1a Street
01-267 Warsaw, Poland
e-mail: tomaszmazurek.wum@gmail.com
tel.: + 48 22 599 19 58
fax.: + 48 22 599 19 57

Reviewer 3 Report

The manuscript is interesting and quite well written. The conclusions are supported by the results. The limitations of the study were pointed out by the authors.

This reviewer raises some issues that the authors need to address.

1- In this observational study, a patient population with severe CAD has, as expected, numerous risk factors in addition to coronary artery damage. In fact, the population is elderly (69 years) and 80% hypertensive and dyslipidemic and 30% diabetic, and on average overweight (BMI 28). Indeed, the optimal medical treatment (OMT) described by the authors is not clear because it does not mainly define how many patients during observation achieved and maintained the therapeutic goals of the various CV risk factors. Very recently, mortality and MACEs in a very high CV risk population has been investigated in the NID-2 study (Cardiovasc Diabetol (2021) 20:145. doi: 10.1186/s12933-021-01343-1). NID-2 study originally demonstrated the ability of a multifactorial therapeutic approach to significantly improve above hard outcome, moreover with a long durability of protection, in a randomized multicentric trial. Therefore, the invitation to a optimized therapy must not be limited to the management of CAD alone, but also of the other main risk factors, as part of an integrated and multifactorial therapy, in order to improve the mortality and CV outcome. This very important issue must be adequately commented in the paper.

2- Tight glycemic control has been described as affecting cardiac remodeling and CV outcome in patients with severe CAD, in particular during STEMI, also in non-diabetic patients (Journal of Clinical Endocrinology and Metabolism Volume 97, Issue 3, March 2012, 933-942. doi: 10.1210/jc.2011-2037 - Journal of Diabetes Research, 2018, art. no. 3106056. doi: 10.1155/2018/3106056). This important issue and above references should be included in the discussion and adequately commented.

3- Notably, the role of adiponectin in CAD has recently been well documented (Cardiovasc Diabetol. 2019 Mar 4; 18 (1): 24. doi: 10.1186 / s12933 - 019-0826-0). This issue and this reference should be added in the introduction.

Author Response

Dear

We greatly appreciate you having reviewed our manuscript.

Below we presented our responses to your comments.

  1. “In this observational study, a patient population with severe CAD has, as expected, numerous risk factors in addition to coronary artery damage. In fact, the population is elderly (69 years) and 80% hypertensive and dyslipidemic and 30% diabetic, and on average overweight (BMI 28). Indeed, the optimal medical treatment (OMT) described by the authors is not clear because it does not mainly define how many patients during observation achieved and maintained the therapeutic goals of the various CV risk factors. Very recently, mortality and MACEs in a very high CV risk population has been investigated in the NID-2 study (Cardiovasc Diabetol (2021) 20:145. doi: 10.1186/s12933-021-01343-1). NID-2 study originally demonstrated the ability of a multifactorial therapeutic approach to significantly improve above hard outcome, moreover with a long durability of protection, in a randomized multicentric trial. Therefore, the invitation to a optimized therapy must not be limited to the management of CAD alone, but also of the other main risk factors, as part of an integrated and multifactorial therapy, in order to improve the mortality and CV outcome. This very important issue must be adequately commented in the paper.”

Our answer:
In section Methods and study design, page 3; line 100-102 we have written: “OMT was defined as using of drugs with proven impact on increased survival or providing optimal reduction of the signs and symptoms associated with CAD.” The types of these drugs used in CABG, PCI and OMT-groups at discharge was included in Table 2. Cardiac-related medications given at discharge.

We agree that we cannot objectively assess the effectiveness of OMT during follow-up, but in this study we did not carry out follow-up visits and we did not perform laboratory tests during follow-up to answer the question of whether the therapeutic goals have been achieved. We only assess endpoints and quality of life at the end of follow-up. We have described this limitation in section Limitations, page 12; line 397-399: Additionally, proper and regular use of drugs by patients often remains a matter of trust, hence it is difficult to determine the credibility of the endpoints in the OMT-group.

In section Limitations, page 12; line 398 we have added: “ during follow-up”.

We agree that to obtain the best treatment results and endpoints, not only adherence to CAD-medications is required, hence in Section Methods and study design, page 3, line 102-104 we have added: “ as well as drugs that optimally control comorbidities accelerating the progression of atherosclerosis and CAD, mainly antihypertensive, antidiabetic and lipid-lowering medications.”

  1. “Tight glycemic control has been described as affecting cardiac remodeling and CV outcome in patients with severe CAD, in particular during STEMI, also in non-diabetic patients (Journal of Clinical Endocrinology and Metabolism Volume 97, Issue 3, March 2012, 933-942. doi: 10.1210/jc.2011-2037 - Journal of Diabetes Research, 2018, art. no. 3106056. doi: 10.1155/2018/3106056). This important issue and above references should be included in the discussion and adequately commented.”

Our answer:

In section Discussion, page 11-12, line 361-372, we have added: “In our study population, 30.6 % of patients had presented with diabetes. It has been widely known that hyperglycemia, prediabetes and diabetes mellitus are the most serious factors responsible for progression of atherosclerosis. Therefore, it should be emphasized that the importance of proper glycemic control in all study groups – CABG, PCI and OMT is one of the most important element improving the prognosis and survival in CAD. Hyperglycemia induces oxidative stress in myocardial damage during acute ischemia and thus reduces potential regenerative abilities of myocardium. When a tight glycemic control is implemented early in peri-ischemia period, it may improve outcomes in ACS, also in non-diabetic patients. That is why MHT cooperation is so important, because multi-specialists team, apart from interventional treatment, can recommend advices on how strictly control factors responsible for development and progression of CAD, which may reflect to the effectiveness of OMT. [24]”

Additionally, in section References, page 14, line 503-506, we have added adequate reference: “ 24. Marfella R, Sasso FC, Cacciapuoti F, Portoghese M, Rizzo MR, Siniscalchi M, Carbonara O, Ferraraccio F, Torella M, Petrella A, Balestrieri ML, Stiuso P, Nappi G, Paolisso G. Tight Glycemic Control May Increase Regenerative Potential of Myocardium during Acute Infarction. J Clin Endocrinol Metab. 2012; 97(3): 933-942. DOI: 10.1210/jc.2011-2037.”

  1. “Notably, the role of adiponectin in CAD has recently been well documented (Cardiovasc Diabetol. 2019 Mar 4; 18 (1): 24. doi: 10.1186 / s12933 - 019-0826-0). This issue and this reference should be added in the introduction.”

Our answer:

In section Introduction, page 2, line 67-73, we have added: “Moreover, new diagnostic markers have emerged in the diagnosis of CAD, including mostly the role of adiponectin in recent years. In the AIRE study, for 54 patients with Normal Glucose Tolerance (NGT) who underwent coronary revascularization by PCI, adiponectin was proved to be independently associated with de novo ischemic heart disease, restenosis and overall new PCI. [22] Understanding and studying new CAD-markers have added to the complexity of optimal treatment of MVD-subjects.”

Additionally, in section References, page 14, line 495-498, we have added adequate reference: “22. Sasso FC, Pafundi PC, Marfella R, Calabrò P, Piscione F, Furbatto F, Esposito G, Galiero R, Gragnano F, Rinaldi L, Salvtore T, D’Amico M, Adinolfi LE, Sardu C. Adiponectin and insulin resistance are related to restenosis and overall new PCI in subjects with normal glucose tolerance: the prospective AIRE Study. Cardiovasc Diabetol. 2019; 18 (1): 24. DOI: 10.1186/s12933-019-0826-0.”

Thank you for taking the time for review this article.

With best regards

Tomasz Mazurek, MD, PhD

1st Department of Cardiology

Medical University of Warsaw,

Banacha 1a Street
01-267 Warsaw, Poland
e-mail: tomaszmazurek.wum@gmail.com
tel.: + 48 22 599 19 58
fax.: + 48 22 599 19 57

Round 2

Reviewer 3 Report

No further comments.